# Robust Control Allocation for Space Inertial Sensor under Test Mass Release Phase with Overcritical Conditions

**DOI:** 10.3390/s23062881

**Published:** 2023-03-07

**Authors:** Juzheng Zhang, Yu Zhang, Wenjian Tao, Zhenkun Lu, Mingpei Lin

**Affiliations:** 1MOE Key Laboratory of TianQin Mission, TianQin Research Center for Gravitational Physics & School of Physics and Astronomy, Frontiers Science Center for TianQin, Gravitational Wave Research Center of CNSA, Sun Yat-sen University (Zhuhai Campus), Zhuhai 519082, China; 2School of Aeronautics and Astronautics, Sun Yat-sen University, Shenzhen 518107, China

**Keywords:** robust control allocation, drag-free spacecraft, space inertial sensor, system uncertainty, 6-DOF control

## Abstract

This paper proposes a robust control allocation for the capture control of the space inertial sensor’s test mass under overcritical conditions. Uncertainty factors of the test mass control system under the overcritical condition are analyzed first, and a 6-DOF test mass dynamics model with system uncertainty is established. Subsequently, a time-varying weight function is designed to coordinate the allocation of 6-DOF generalized forces. Moreover, a robust control allocation method is proposed to distribute the commanded forces and torques into individual electrodes in an optimal manner, which takes into account the system uncertainties. This method transforms the robust control allocation problem into a second-order cone optimization problem, and its dual problem is introduced to simplify the computational complexity and improve the solving efficiency. Numerical simulation results are presented to illustrate and highlight the fine performance benefits obtained using the proposed robust control allocation method, which improves capture efficiency, increases the security margin and reduces allocation errors.

## 1. Introduction

In recent years, drag-free spacecraft have been successfully used in space missions that require ultra-high stability such as space high-precision earth gravity field surveys [1,2,3], space general-relativity verification [4,5], and space gravitational wave detection [6,7]. The success of these missions has greatly encouraged more scholars to research drag-free control systems. The space inertial sensor (SIS) [8,9,10,11] is one of the key payloads of drag-free spacecraft, which consists of a test mass as an inertial reference and a set of electrodes acting as capacitive sensors and electrostatic actuators. During the launch and the following orbital transfer phase, the test mass is locked inside the electrode house by the caging and vent mechanism (CVM). After the drag-free spacecraft enters the target orbit, the test mass has to be released from the CVM to prepare for science operations [12]. Since the adhesion force and the position of the two thimbles are asymmetrical on both sides of the test mass, the release process will generate the residual speed. During the first in-flight release of the LISA Pathfinder mission, the test mass had a high residual momentum and it was not able to be stabilized until after several undesired impacts took place [13]. As consequence, temporary manual release procedures were adopted. However, the manual procedures cannot be applied to the drag-free spacecraft of deep space exploration missions because the distance between the spacecraft and the Earth produces a large telemetry and telecommand execution time lag. Therefore, it is necessary to design a control system with autonomous stability control capability under overcritical release conditions of test mass.

In the past design of space inertial sensors, the non-redundant configuration of 6 pairs of electrodes controlling the 6-DOF motion of the test mass was generally used [14,15,16], which greatly increase the probability of the single point of failure. For instance, a fault occurred in the position sensing of the test mass of CHAMP spacecraft in the *x*-axis direction, resulting in large errors in the measured data [17]. To improve the fault tolerance and reliability of SIS, the redundant design of electrode configuration is necessary. The redundancy of the electrodes makes the mapping of the commanded electrostatic forces and torques to the control voltage of electrodes not unique. Control allocation is introduced to deal with distributing the total control demand among the individual actuators while accounting for their constraints under redundant actuators.

In the design of control allocation schemes, it is usually assumed that the efficiency matrix is accurately known. However, uncertainty is ubiquitous in practical systems. In the overcritical condition, on the one hand, the collision between the test mass and the electrode housing inevitably makes the test mass carry additional charges. On the other hand, uncalibrated measurement circuits cause measurement errors in the state of test mass and voltage errors in the control of electrodes. Therefore, a robust control allocation method is required that maps the commanded electrostatic force to each electrode even in the presence of uncertainty. In the past ten years, some results have been developed. In [18,19,20], to solve the distributing problem of attitude control torque, robust control allocation was formulated as a min–max optimization problem, which dealt with actuator faults directly without reconfiguring the controller and ensured some robustness of system performances. In [21], the unstructured uncertainty, structured uncertainty, and linear fractional structured uncertainty in the control effectiveness matrix were considered. Robust least-squares allocation methods were, respectively, designed for each kind of uncertainty. In [22], an LMI-based robust control allocation was employed to deal with the problem of distributing the three-axis torques over the reaction wheels. In [23], an online robust control allocation scheme was proposed to redistribute virtual control signals to the actuators taking into account the actuator saturation, uncertainties and faults. The scheme allows a smooth switch between thrusters and reaction wheels, which handles the inaccuracy problem of thrusters and the saturation problem of reaction wheels. In [24], a robust attainable moment set was designed as the worst-case moment set for a given bounded uncertainty caused by variations in flight conditions, vehicle configuration and parametric uncertainties. In [25], to distribute the virtual control law to each actuator in an optimal manner with fault-tolerant ability, a robust control allocation algorithm was developed based on the virtual controller and the iterative learning observer.

The existing robust control allocation methods have the following shortcomings in solving the problem of capture control of the space inertial sensor’s test mass under overcritical conditions. Firstly, the existing researches on robust control allocation usually focus on the allocation of control torque of attitude control system, while the control allocation problem of test mass control allocates force and torque at the same time. Insufficient investigations are carried out on the 6-DOF robust control allocation. Additionally, in the previous research on robust control allocation, the allocation error and energy consumption are usually taken as the optimization objectives. However, under overcritical conditions, avoiding the collision between the test mass and the electrode house is one of the main objectives. There is a lack of robust control allocation methods for such complex task objectives.

To solve the problem of allocation of space inertial sensor under test mass release phase with overcritical conditions, a novel 6-DOF robust control allocation considering both measurement uncertainty and actuation uncertainty is proposed in this paper. The main contributions of this paper are summarized as follows.

(1) The test mass control system considering the state measurement uncertainty and control voltage uncertainty is established.

(2) A time-varying weight function is designed in the optimization function to coordinate the allocation of forces and torques and avoid the collision between the test mass and the electrode house.

(3) The robust control allocation problem is formulated as a second-order cone programming problem, and its dual problem is introduced to improve the solving efficiency.

The paper is structured as follows. System dynamics modeling with uncertainty is given in Section 2. The proposed 6-DOF robust control allocation method is given in Section 3. In Section 4, the application of the proposed robust control allocation is illustrated by several numerical simulations. Conclusions are given in Section 5.

## 2. System Dynamics Modeling with Uncertainty

### 2.1. Relative Dynamics of Spacecraft and Test Mass

Relative dynamics of spacecraft and test mass is the foundation of the control system design. To describe the mathematical model, several coordinate frames are defined at first (seen in Figure 1).

♦ECI (*EX_E_Y_E_Z_E_*) frame: the Earth-centered inertial frame that is centered at the Earth.♦SBF (*O_S_X_S_Y_S_Z_S_*) frame: the spacecraft body-fixed frame and its origin sits at the mass center of the spacecraft.♦TBF (*O_T_X_T_Y_T_Z_T_*) frame: the test mass body-fixed frame and its origin sits at the mass center of the test mass.♦IBF (*O_I_X_I_Y_I_Z_I_*) frame: the space inertial sensor body-fixed frame and its origin sits at the center of the space inertial sensor.

The relative motion of spacecraft and space inertial sensor’s test mass in the IBF frame have been modeled by [26] as follows:(1)$$\left[\begin{array}{c} \dot{\rho }\\ \dot{\zeta } \end{array}\right]=\left[\begin{array}{cc} A_{\rho } & 0\\ 0 & A_{\zeta } \end{array}\right]\left[\begin{array}{c} \rho \\ \zeta \end{array}\right]+\left[\begin{array}{c} -B_{\rho }\\ B_{\zeta } \end{array}\right]+\left[\begin{array}{c} D_{\rho }\rho_{IS}\\ 0 \end{array}\right]+\left[\begin{array}{cc} \frac{1}{m} & 0\\ 0 & \frac{1}{J} \end{array}\right]\left[\begin{array}{c} C\left(F+f\right)\\ C\left(M+T\right) \end{array}\right],\;$$
where $\rho ={\left[x,\dot{x},y,\dot{y},z,\dot{z}\right]}^{T}$, $\zeta ={\left[\varphi ,\dot{\varphi },\theta ,\dot{\theta },\psi ,\dot{\psi }\right]}^{T}$, $\rho_{IS}={\left[\rho_{ISx},\rho_{ISy},\rho_{ISz}\right]}^{T}$,
$$A_{\rho }=\left[\begin{array}{cccccc} 0 & 1 & 0 & 0 & 0 & 0\\ -\frac{\mu }{r_{ETM}^{3}}+\omega_{0}^{2} & 0 & 0 & 0 & -{\dot{\omega }}_{0} & -2\omega_{0}\\ 0 & 0 & 0 & 1 & 0 & 0\\ 0 & 0 & -\frac{\mu }{r_{ETM}^{3}} & 0 & 0 & 0\\ 0 & 0 & 0 & 0 & 0 & 1\\ {\dot{\omega }}_{0} & 2\omega_{0} & 0 & 0 & -\frac{\mu }{r_{ETM}^{3}}+\omega_{0}^{2} & 0 \end{array}\right],$$
$$A_{\zeta }=\left[\begin{array}{cccccc} 0 & 1 & 0 & 0 & 0 & 0\\ -{\dot{\omega }}_{0}\sin\alpha_{SI} & -\omega_{0}\sin\alpha_{SI} & 0 & 0 & -{\dot{\omega }}_{0}\cos\alpha_{SI} & -\omega_{0}\cos\alpha_{SI}\\ 0 & 0 & 0 & 1 & 0 & 0\\ 0 & 0 & 0 & 0 & 0 & 0\\ 0 & 0 & 0 & 0 & 0 & 1\\ {\dot{\omega }}_{0}\cos\alpha_{SI} & \omega_{0}\cos\alpha_{SI} & 0 & 0 & -{\dot{\omega }}_{0}\sin\alpha_{SI} & -\omega_{0}\sin\alpha_{SI} \end{array}\right],$$
$$B_{\rho }=\frac{\mu }{r_{ETM}^{3}}r_{ES}{\left[0,\sin\alpha_{SI},0,0,0,-\cos\alpha_{SI}\right]}^{T},$$
$$B_{\zeta }={\left[0,\theta \psi {\dot{\omega }}_{0}\sin\alpha_{SI},0,-{\dot{\omega }}_{0},0,-\theta \psi {\dot{\omega }}_{0}\cos\alpha_{SI}\right]}^{T},$$
$$\begin{array}{l} D_{\rho }=\left[\begin{array}{ccc} -\cos\alpha_{SI} & 0 & \sin\alpha_{SI}\\ -(\frac{\mu }{r_{ETM}^{3}}+\omega_{0}^{2})\cos\alpha_{SI}-{\dot{\omega }}_{0}\sin\alpha_{SI} & 0 & -(-\frac{\mu }{r_{ETM}^{3}}+\omega_{0}^{2})\sin\alpha_{SI}-{\dot{\omega }}_{0}\cos\alpha_{SI}\\ 0 & -1 & 0\\ 0 & -\frac{\mu }{r_{ETM}^{3}} & 0\\ -\sin\alpha_{SI} & 0 & -\cos\alpha_{SI}\\ (-\frac{\mu }{r_{ETM}^{3}}+\omega_{0}^{2})\sin\alpha_{SI}+{\dot{\omega }}_{0}\cos\alpha_{SI} & 0 & -(\frac{\mu }{r_{ETM}^{3}}+\omega_{0}^{2})\cos\alpha_{SI}-{\dot{\omega }}_{0}\sin\alpha_{SI} \end{array}\right] \end{array}$$
$$C={\left[\begin{array}{cccccc} 0 & 1 & 0 & 0 & 0 & 0\\ 0 & 0 & 0 & 1 & 0 & 0\\ 0 & 0 & 0 & 0 & 0 & 1 \end{array}\right]}^{T}.$$
where $r={\left(x,y,z\right)}^{T}$ and ${\left(\dot{x},\dot{y},\dot{z}\right)}^{T}$ denote the position vector and the velocity vector of the test mass in IBF frame, respectively. ${\left(\varphi ,\theta ,\psi \right)}^{T}$ and ${\left(\dot{\varphi },\dot{\theta },\dot{\psi }\right)}^{T}$ denote the attitude angle and the attitude angular velocity of test mass in IBF frame, respectively. ${\left(\rho_{ISx},\rho_{ISy},\rho_{ISz}\right)}^{T}$ is the position vector of the inertial senser center relative to the spacecraft mass center. $\alpha_{SI}$ is a rotational angle from SBF frame to IBF frame and it is a constant. μ is Earth’s gravitational constant. $r_{ETM}$ and $r_{ES}$ are the distance of test mass and spacecraft from the Earth, respectively. $\omega_{0}$ and ${\dot{\omega }}_{0}$ are the angular velocity and angular acceleration of the SBF frame relative to ECI frame.

### 2.2. Electrostatic Force Model

Inside the spacecraft, surrounding the test mass, are a set of electrodes, forming a capacitive space inertial sensor, for sensing and actuating the state of the test mass. The mathematical model of the electrostatic forces/torques can be derived from standard energetic considerations, which can be written as [27]
(2)$$\left\{\begin{array}{c} F_{q}=\frac{1}{2}\overset{n}{\underset{i=1}{\sum }}\frac{\partial C_{i,TM}}{\partial q}{\left(V_{i}-V_{TM}\right)}^{2}\\ V_{TM}=\frac{Q+\sum_{i=1}^{n}C_{i,TM}V_{i}}{\sum_{i=1}^{n}C_{i,TM}}, \end{array}\right.$$
where $C_{i,TM}$ denotes the capacitance of the *i*-th electrodes and the test mass, $V_{i}$ denotes the potential on the *i*-th electrodes, $V_{TM}$ denotes the potential on the test mass and *q* is the generalized coordinate, and *Q* denotes the charge on the test mass.

Ignoring the fridge effect, the capacitance between two parallel electrodes is: (3)$$C_{0}=\frac{\varepsilon_{0}\varepsilon_{r}A}{d_{0}},$$
where $\varepsilon_{0}$ and $\varepsilon_{r}$ are the vacuum permittivity and the relative permittivity, respectively. A is the area of the electrode, and *d*_0_ is the distance between the electrodes.

When the distance between the centers of electrodes does not change, and one of the electrodes is rotated around the *x*-axis and *y*-axis of the center of the plate by angles α and β, as shown in Figure 2. The capacitance between the electrodes can be calculated by employing second-order approximation technique based on Taylor expansion as follows:


(4)
$$\begin{array}{cc} C_{1}= & \underset{-\frac{W}{2}\leq x\leq \frac{W}{2},-\frac{L}{2}\leq y\leq \frac{L}{2}}{\int \int }\frac{\varepsilon dxdy}{d_{0}-x\beta +y\alpha }\\ \approx & \underset{-\frac{W}{2}\leq x\leq \frac{W}{2},-\frac{L}{2}\leq y\leq \frac{L}{2}}{\int \int }\varepsilon \left(\frac{1}{d_{0}}+\frac{x\beta }{d_{0}^{2}}-\frac{y\alpha }{d_{0}^{2}}+\frac{x^{2}\beta^{2}}{d_{0}^{3}}+\frac{y^{2}\alpha^{2}}{d_{0}^{3}}-\frac{2x\beta y\alpha }{d_{0}^{3}}\right)dxdy\\ \approx & \underset{-\frac{W}{2}\leq x\leq \frac{W}{2},-\frac{L}{2}\leq y\leq \frac{L}{2}}{\int \int }\varepsilon \left(\frac{1}{d_{0}}+\frac{x\beta }{d_{0}^{2}}-\frac{y\alpha }{d_{0}^{2}}\right)=\frac{\varepsilon A}{d}+\frac{\varepsilon A\left(L^{2}\alpha^{2}+W^{2}\beta^{2}\right)}{12d_{0}^{3}} \end{array}$$


Comparing Equations (3) and (4), it can be known that the difference between the two formulas is the second-order small quantity of relative rotation angles, which is too small to neglect. Therefore, it can be considered that the capacitance between two electrodes with relative rotation is only related to the center distance of the electrodes, which can be written as
(5)$$C_{i}=\frac{\varepsilon A_{i}}{d_{i}},$$
where *A_i_* and *d_i_* are the area of the *i*-th electrodes and the distance of the electrodes to the surface of the test mass, respectively.

The transformation matrix from TBF frame to IBF matrix can be written as
(6)$$T_{IT}=\left[\begin{array}{ccc} C\theta C\psi -S\varphi S\theta S\psi & C\theta S\psi +S\varphi S\theta C\psi & -C\varphi S\theta \\ -C\varphi S\psi & C\varphi C\psi & S\varphi \\ S\theta C\psi +S\varphi C\theta S\psi & S\theta S\psi -S\varphi C\theta C\psi & C\varphi C\theta \end{array}\right],\;$$
where *C* [∙] and *S* [∙] are simplified notation for *cos* [∙] and *sin* [∙]. Considering the small attitude angle, the sine and cosine functions can be approximated as sin ε≈ε, cos ε≈1. Accordingly, Equation (6) could be simplified as
(7)$$T_{IT}\approx \left[\begin{array}{ccc} 1 & \psi & -\theta \\ -\psi & 1 & \varphi \\ \theta & -\varphi & 1 \end{array}\right].\;$$

After releasing, the distance variation of the test mass and electrode consists of the translation of the center of mass and the rotation about the center of mass. The distance from the *i*-th electrode to the test mass is
(8)$$d_{i}=d_{0i}+\left[(T_{IT}-I_{3})P_{Tmi}+r\right]t_{0i},\;$$
where $P_{Tmi}$ is the position vector of the $P_{i}$ in the TBF frame, and $P_{i}$ is the point which is on the test mass’s surface and opposites the center of the *i*-th electrode before releasing. $t_{0i}$ is the unit vector from the point $P_{i}$ to the center of the *i*-th electrode. *d*_0*i*_ is denoted as the initial distance between the point $P_{i}$ to the center of the *i*-th electrode.

By substituting Equations (5) and (8) into Equation (2), and choosing the generalized coordinate as 3-DOF translation and 3-DOF rotation, the nominal generalized electrostatic force model is given as follows:
(9)v=εA1∂d1−1∂x⋯εAn∂dn−1∂xεA1∂d1−1∂y⋯εAn∂dn−1∂yεA1∂d1−1∂z⋯εAn∂dn−1∂zεA1∂d1−1∂φ⋯εAn∂dn−1∂φεA1∂d1−1∂θ⋯εAn∂dn−1∂θεA1∂d1−1∂ψ⋯εAn∂dn−1∂ψ(V1−VTM)2(V2−VTM)2⋯(Vn−VTM)2=∂D∂qU=BU, where ***v*** is the generalized electrostatic force, D=εA1d1−1,A2d2−1,⋯Andn−1∈ℝ1×n, q∈ℝ6×1 denotes the generalized degrees of freedom, B∈ℝ6×n denotes the control efficiency matrix, and U∈ℝn×1 denote the square of the potential difference between the electrode and the test mass.

### 2.3. Uncertainties in Control System

It is well known that uncertainty in control systems is usually divided into two categories: measure uncertainty and actuator uncertainty. In the test mass control system, on the one hand, the position and attitude of test mass in IBF frame are determined from the capacitance that measured by electrodes [28]. During the test mass capture phase, the capacitance measurement circuit has not been calibrated and there existed measurement error. The measured distance can be written as ${\tilde{d}}_{i}=d_{i}+\Delta d_{i}$, where $\Delta d_{i}$ denote the measurement error of $d_{i}$. Then, ignoring the higher order of the measurement error, the ***B***(*i*, *j*) can be written as: (10)$$\tilde{B}(i,j)=\frac{\partial {\tilde{D}}_{i}}{\partial q_{j}}=\varepsilon A_{j}\frac{\partial {\left[(d_{i}+\Delta d_{i})\right]}^{-1}}{\partial q_{j}}=\varepsilon A_{j}\frac{\partial [(d_{i}^{-1}-d_{i}^{-2}\Delta d_{i}+o(\Delta d_{i})]}{\partial q_{j}}\approx \varepsilon A_{j}\left[\frac{\partial d_{i}{}^{-1}}{\partial q_{j}}-\frac{\partial (d_{i}^{-2}\Delta d_{i})}{\partial q_{j}}\right].$$

On the other hand, the charge management system has not been able to control the charge of the test mass to an ideal state, and collision with the electrode housing also inevitably makes the test mass carry additional charges [29]. Therefore, the square of the potential difference in Equation (9) with uncertainty can be written as
(11)$${\tilde{U}}_{i}=U_{i}+\Delta U_{i}.\;$$

According to Equations (10) and (11), the actual generalized electrostatic force can be written as
(12)v=(B+ΔB)(U+ΔU). 


**Assumption** **1.**
*the uncertain control efficiency matrix *

ΔB

*, and the uncertainty of the square of the potential difference vector *

ΔU

* are assumed to be unknown but bounded, i.e., there exist constants η and σ, respectively, to satisfying:*


(13)
$$\left\{\begin{array}{c} \left\|\Delta B\right\|\leq \eta \\ \left\|\Delta U\right\|\leq \sigma \end{array}\right..$$



## 3. 6-DOF Robust Control Allocation

A control system with redundant actuators can usually be divided into two serial levels, which can be seen in Figure 3. Firstly, a motion control algorithm is designed as a high-level controller that takes reference signals ($\left[\rho_{d}\;\zeta_{d}\right]$) and system states as inputs and outputs the commanded forces and torques. Secondly, a control allocation algorithm is designed to map the commanded forces and torques to the redundant actuators. The mapping scheme is not unique, which enables secondary control objectives such as power minimization and fuel equalization to be achieved on the basis of satisfying the commanded forces and torques. Independent design of the motion control algorithm and control allocation algorithm greatly reduces the complexity of the control system. The motion control algorithm ensures the control performance such as the stability of the system, and the control allocation algorithm handles problems such as actuator saturation and failure.

To minimize the worst-case allocation residual under control constraints, a robust control allocation problem with uncertainties can be described as an optimization problem with constraints: (14)$$U=\arg\underset{\underline{U}\leq U\leq \bar{U}}{\min}\underset{\left\|\Delta B\right\|\leq \eta ,\left\|\Delta U\right\|\leq \sigma }{\max}\left[\left\|W_{U}U\right\|+\gamma \left\|W_{v}\left(\left(B+\Delta B\right)\left(U+\Delta U\right)-v_{d}\right)\right\|\right],\;$$
where ***W_U_*** is a positive definite matrix and it is used to reduce the magnitude of ***U*** and ensure the uniqueness of the solution. *γ* is a positive constant that indicates the weight of the allocation residual. ***v_d_*** is the desired generalized electrostatic force. ***Wv*** is the positive definite matrix that indicates the weight of allocation residual for the generalized electrostatic force of each degree of freedom.

### 3.1. Weight Function Design

The test mass is a convex geometry. When it collides with the electrode house, one of the vertices of the test mass will contact with the electrode house at the first time. Therefore, in order to avoid collisions as much as possible, the ***Wv*** matrix uses vertices position and test mass state to design a weight matrix.

Denoting $r_{j}^{TBF}$ as the position vector of the *j*-th vertex of test mass in the TBF frame, after the test mass is released, the position vector in IBF frame is
(15)$$r_{j}^{IBF}=T_{IT}r_{j}^{TBF}+r\;(j=1,2,\cdots ,8).\;$$

The position vectors of the center of the electrode house surface that are adjacent to the *j*-th vertex in the IBF system are denoted as (*x_s_*, 0, 0)*^T^*, (0, *y_s_*, 0)*^T^*, (0, 0, *z_s_*)*^T^*. The vector from the *j*-th vertex to the three surfaces is
(16)$$p_{j}=\left[\begin{array}{c} p_{x}\\ p_{y}\\ p_{z} \end{array}\right]=\left[\begin{array}{c} x_{s}\\ y_{s}\\ z_{s} \end{array}\right]-r_{j}^{IBF}\;(j=1,2,\cdots ,8).$$

The velocity of the *j*-th vertex moving to each surface of the surfaces consists of the translation of the center of mass and the rotation about the center of mass. The translation velocity is $v_{tj}=v_{Tm}$, and the velocity that rotation about the center of mass is $v_{rj}={\left(r_{j}^{TBF}\right)}_{\times }\omega_{TM}$, where ${\left(\cdot \right)}_{\times }$ is the cross-product operator since $a\times b=a^{\times }b$, and $\omega_{TM}={\left(\dot{\varphi },\dot{\theta },\dot{\psi }\right)}^{T}$.

To characterize the risk of collision, the auxiliary parameters are defined as
(17)$$\left[\begin{array}{c} \xi_{xj}\\ \xi_{yj}\\ \xi_{zj} \end{array}\right]=diag\left(v_{TM}\right)\cdot diag{\left(p_{j}\right)}^{-1}\cdot e\left(3\right),$$
(18)$$\begin{array}{c} \;\left[\begin{array}{c} \xi_{\varphi j}\\ \xi_{\theta j}\\ \xi_{\psi j} \end{array}\right]=diag\left(v_{rj}\right)\cdot diag{\left(p_{j}\right)}^{-1}\cdot e\left(3\right) \end{array},\;$$
where $\xi_{ij}$ represents the reciprocal of the remaining time for collision at the *j*-th vertex only under the action of velocity or angular velocity of *i*-th DOF, and $e\left(3\right)={\left[\begin{array}{ccc} 1 & 1 & 1 \end{array}\right]}^{T}$. The negative value of $\xi_{ij}$ indicates that the vertex is away from the electrode house, so all negative values are set to zero as
(19)$$\xi_{i}^{p}=\left\{\begin{array}{cc} \xi_{ij} & if\;\xi_{ij}>0\\ 0 & if\;\xi_{ij}\leq 0 \end{array}\right..$$

The auxiliary parameter of *i*-th DOF is defined as
(20)$$\xi_{i}=\overset{8}{\underset{j=1}{\sum }}\xi_{ij}^{p}+\kappa ,\;$$
where κ is a constant used to ensure that the force or torque of the *i*-th DOF can be well allocated even if the risk of collision in *i*-th DOF is minimal.

The weighted matrix is designed as follows: (21)$$\begin{array}{c} W_{v1}=diag\left[th\left(\xi_{x}\right),th\left(\xi_{y}\right),th\left(\xi_{z}\right),th\left(\xi_{\varphi }\right),th\left(\xi_{\theta }\right),th\left(\xi_{\psi }\right)\right] \end{array},\;$$
where *th*(·) is the hyperbolic tangent function since $th\left(k\right)=\frac{e^{k}-e^{-k}}{e^{k}+e^{-k}}$.

In order to unify the forces and torques to the magnitude of force, the weight matrix is given as
(22)$$W_{v2}=diag\left(1,1,1,\frac{1}{L_{\varphi }},\frac{1}{L_{\theta }},\frac{1}{L_{\psi }}\right),\;$$
where *L_i_* is the moment arm of the electrostatic moment of the *i*-th degree of freedom.

Combining Equation (21) and Equation (22), the adaptive weight matrix is designed as
(23)$$W_{v}=\frac{1}{d}W_{v1}\cdot W_{v2},\;$$
where $d=norm\left(W_{v1}\right)\cdot norm\left(W_{v2}\right)$, so the norm of ***W_v_*** is one.

### 3.2. Robust Control Allocation Design

The robust control allocation problem described in Equation (14) can be rewritten as follows: (24)$$U=\arg\underset{\underline{U}\leq U\leq \bar{U}}{\min}\left\{\left\|W_{U}U\right\|+\underset{\left\|\Delta B\right\|\leq \eta ,\left\|\Delta U\right\|\leq \sigma }{\max}\gamma \left\|W_{v}\left[\left(BU-v_{d}\right)+\Delta BU+B\Delta U+\Delta B\Delta U\right]\right\|\right\}.$$

To solve the min–max problem, the inner maximization problem is solved first. Denoting y=BΔU+ΔBΔU, and $\Delta y=\left(\left\|B\right\|+\left\|\Delta B\right\|\right)\left\|\Delta U\right\|=\left(\left\|B\right\|+\eta \right)\sigma$. According to the triangle inequality, $\left\|y\right\|=\left\|B+\Delta B\right\|\left\|\Delta U\right\|\leq \Delta y$. Ulteriorly, let z(U)=ΔBU+Δy, and $z(U)=\left\|\Delta B\right\|\left\|\Delta U\right\|+\Delta y=\eta U+(\left\|B\right\|+\eta )\sigma$.

The robust control allocation problem can be rewritten as
(25)$$U=\arg\underset{\underline{U}\leq U\leq \bar{U}}{\min}\left(\left\|W_{U}U\right\|+\underset{\left\|z\right\|\leq z(U)}{\max}\gamma \left\|W_{v}\left[BU-v_{d}+z\right]\right\|\right).\;$$

The second term in brackets of Equation (25) is convex in ***U***, and the constraint **z**(***U***) is also convex in ***U***. Therefore, the maximum of the second term is obtained at the boundary of z(U). Using the triangle inequality: (26)$$\begin{array}{c} \left\|W_{v}(BU-v_{d}+z)\right\|\leq \left\|W_{v}(BU-v_{d})\right\|+\left\|W_{v}z\right\|\\ \;\;\;\;\;\;\;\;\;\;\;\;\;\;\;\;\;\;\;\leq \left\|W_{v}(BU-v_{d})\right\|+\eta \left\|W_{v}\right\|\left\|U\right\|+\left\|W_{v}\right\|\left(\left\|B\right\|+\eta \right)\sigma \\ \;\;\;\;\;\;\;\;\;\;\;\;\;=\left\|W_{v}(BU-v_{d})\right\|+\eta \left\|U\right\|+\left(\left\|B\right\|+\eta \right)\sigma . \end{array}$$

The upper bound in Equation (26) is obtained when
(27)$$z=\varsigma \left[\eta \left\|U\right\|+\left(\left\|B\right\|+\eta \right)\sigma \right],$$
(28)$$\varsigma =\left\{\begin{array}{cc} \frac{\left(BU-v_{d}\right)}{\left(BU-v_{d}\right)} & if\;BU\neq v_{d}\\ \mathrm{any}\;\mathrm{unit}\;\mathrm{norm}\;\mathrm{vector} & if\;BU=v_{d}. \end{array}\right.$$
(29)$$\begin{array}{c} U=\arg\underset{\underline{U}\leq U\leq \bar{U}}{\min}\left\{\left\|W_{U}U\right\|+\gamma \left[\left\|W_{v}\left(BU-v_{d}\right)\right\|+\left\|\eta U\right\|+\left(\left\|B\right\|+\eta \right)\sigma \right]\right\}\\ \;=\arg\underset{\underline{U}\leq U\leq \bar{U}}{\min}\gamma \left\{\left\|W_{v}\left(BU-v_{d}\right)\right\|+\rho \left\|U\right\|+\tau \right\}, \end{array}$$
where $\rho =\eta +\left\|W_{U}\right\|/\gamma$, and $\tau =\left(\left\|B\right\|+\eta \right)\sigma$.

The Equation (29) can be rewritten as a second-order cone programming problem formulation as follows: (30)$$\begin{array}{c} \underset{U,\;t_{1},\;t_{2}}{\min}t_{1}+t_{2}\\ s.t.\left\{\begin{array}{c} \left\|W_{v}(BU-v_{d})\right\|\leq t_{1}\\ \left\|U\right\|\leq \frac{t_{2}}{\rho }\\ \underline{U}\leq U\leq \bar{U}. \end{array}\right. \end{array}$$

If $BU=v_{d}$, the problem described in Equation (30) degenerates into the quadratic optimal problem with linear constraints, and the solution is obtained by pseudo-inverse method $U=B^{T}{\left(BB^{T}\right)}^{-1}v_{d}$.

If $v_{d}>BU$, defining the Lagrangian ***L*** associated with the problem (30) as
(31)$$L\left(U,k_{1},k_{2},k_{3},k_{4}\right)=k_{1}^{T}W_{v}\left(BU-v_{d}\right)+k_{2}^{T}U+k_{3}^{T}\left(\underline{U}-U\right)+k_{4}^{T}\left(U-\bar{U}\right).$$

Denote the optimal value of the Equation (30) as $p^{*}$, it can be obtained by the following min–max function: (32)$$p^{*}=\underset{U}{\min}\underset{\begin{array}{c} \left\|k_{1}\right\|\leq 1,\left\|k_{2}\right\|\leq \rho \\ k_{3},k_{4}\geq 0 \end{array}}{\max}L\left(U,k_{1},k_{2},k_{3},k_{4}\right).\;$$

Exchanging the minimize function and the maximum function leads to the dual problem as
(33)$$d^{*}=\underset{\begin{array}{c} \left\|k_{1}\right\|\leq 1,\left\|k_{2}\right\|\leq \rho \\ k_{3},k_{4}\geq 0 \end{array}}{\max}\underset{U}{\min}L\left(U,k_{1},k_{2},k_{3},k_{4}\right).\;$$

The Lagrange dual function is defined as follows: (34)$$\begin{array}{c} g\left(k_{1},k_{2},k_{3},k_{4}\right)=\underset{U}{\inf}L\left(U,k_{1},k_{2},k_{3},k_{4}\right)\\ \;\;\;\;\;\;\;\;\;\;\;\;\;\;\;\;\;\;\;\;=\underset{U}{\inf}k_{1}^{T}W_{v}\left(BU-v_{d}\right)+k_{2}^{T}U+k_{3}^{T}\left(\underline{U}-U\right)+k_{4}^{T}\left(U-\bar{U}\right)\\ \;\;\;\;\;\;\;\;\;\;\;\;\;\;\;\;\;\;\;\;\;\;\;\;\;\;=\left\{\begin{array}{cc} -k_{1}^{T}W_{v}v_{d}+k_{3}^{T}\underline{U}-k_{4}^{T}\bar{U} & if\;k_{1}^{T}W_{v}B+k_{2}^{T}-k_{3}^{T}+k_{4}^{T}=0\\ -\infty & \mathrm{otherwise} \end{array}\right., \end{array}$$
where inf is abbreviation of infimum and the function *g* is the infimum of a family of linear functions of ***U***.

Therefore, the dual problem of Equation (30) is obtained: (35)$$\begin{array}{l} \underset{k_{1},k_{2},k_{3},k_{4}}{\max}-k_{1}^{T}W_{v}v_{d}+k_{3}^{T}\underline{U}-k_{4}^{T}\bar{U}\\ s.t.\left\{\begin{array}{c} \left\|k_{1}\right\|\leq 1\\ \left\|k_{2}\right\|\leq \rho \\ k_{3}\geq 0\\ k_{4}\geq 0\\ k_{1}^{T}W_{v}B+k_{2}^{T}-k_{3}^{T}+k_{4}^{T}=0. \end{array}\right. \end{array}$$

It can be known from Equation (31) and the inner maximization problem of Equation (32) that the maximal value is obtained if and only if
(36)$$k_{1}=\frac{W_{v}\left(BU-v_{d}\right)}{\left\|W_{v}\left(BU-v_{d}\right)\right\|},\;$$
(37)$$k_{2}=\rho \frac{U}{\left\|U\right\|}.\;$$

For the above second-order cone programming, the optimal value $p^{*}$ is bounded and finite according to the physical meaning. Thus, strong duality holds and we have [30]
(38)$$p^{*}=d^{*}.$$

Substituting Equations (36) and (37) into Equation (35) gives
(39)$$\begin{array}{c} \left\|W_{v}\left(BU-v_{d}\right)\right\|+\rho \left\|U\right\|=-k_{1}^{T}W_{v}v_{d}+k_{3}^{T}\underline{U}-k_{4}^{T}\bar{U}\\ \;\;\;\;\;\;\;\;\;\;\;\;\;\;\;\;\;\;=k_{1}^{T}W_{v}\left(BU-v_{d}\right)-k_{1}^{T}W_{v}BU+k_{3}^{T}\underline{U}-k_{4}^{T}\bar{U}, \end{array}$$
(40)$$\rho \left\|U\right\|=-k_{1}^{T}W_{v}BU+k_{3}^{T}\underline{U}-k_{4}^{T}\bar{U}=\left[U^{T}\;{\underline{U}}^{T}\;{\bar{U}}^{T}\right]\left[\begin{array}{c} -k_{1}^{T}W_{v}B\\ k_{3}\\ -k_{4} \end{array}\right].\;$$

The pseudo-inverse solution of Equation (40) is
(41)$$\left[\begin{array}{c} -k_{1}^{T}W_{v}B\\ k_{2}\\ k_{3} \end{array}\right]=\frac{\rho \left\|U\right\|}{{\left\|U\right\|}^{2}+{\left\|\underline{U}\right\|}^{2}+{\left\|\bar{U}\right\|}^{2}}\left[\begin{array}{c} U\\ \underline{U}\\ \dot{U} \end{array}\right]$$

By substituting Equation (36) into Equation (41), the solution of the robust control allocation problem is yielded: (42)$$\begin{array}{c} U_{R\mathrm{CA}}={\left(\frac{B^{T}W_{v}^{T}W_{v}B}{\left\|W_{v}\left(BU-v_{d}\right)\right\|}+\frac{\rho \left\|U\right\|}{{\left\|U\right\|}^{2}+{\left\|\underline{U}\right\|}^{2}+{\left\|\bar{U}\right\|}^{2}}I_{n}\right)}^{-1}\frac{B^{T}W_{v}^{T}v_{d}}{\left\|W_{v}\left(BU-v_{d}\right)\right\|}\\ \;\;\;\;\;={\left[B^{T}W_{v}^{T}W_{v}B+\frac{t_{1}t_{2}}{{\left(\frac{t_{2}}{\rho }\right)}^{2}+{\left\|\underline{U}\right\|}^{2}+{\left\|\bar{U}\right\|}^{2}}I_{n}\right]}^{-1}B^{T}W_{v}^{T}W_{v}v_{d}. \end{array}$$

It should be noted that the solution of Equation (42) is the square of the potential difference between the electrodes and the test mass. To avoid cross-axis coupling effects, the potential of the test mass is controlled to approach zero. In order to reduce the electric potential of the test mass, an additional objective is to let $\overset{n}{\underset{i=1}{\sum }}C_{i,\mathrm{TM}}V_{i}$ near zero. The potential of electrodes *V_i_* can be divided into two groups, one group has negative values and the other group has non-negative values, the two groups would cancel out each other and result in the minimization of $\left|\overset{n}{\underset{i=1}{\sum }}C_{i,\mathrm{TM}}V_{i}\right|$. To divide $C_{i,\mathrm{TM}}V_{i}$ into two groups as equal as possible, the problem is transformed into 0–1 knapsack problem:(43)$$\begin{array}{c} \max\;\overset{n}{\underset{i=1}{\sum }}f_{i}C_{i,TM}V_{pi}\\ s.t.\left\{\begin{array}{c} \overset{n}{\underset{i=1}{\sum }}f_{i}C_{i,TM}V_{pi}\leq \frac{C_{i}V_{pi}}{2}\\ f_{i}\in \left\{0,1\right\}, \end{array}\right. \end{array}$$
where $C_{i,TM}$ can be obtained by Equations (5) and (8), $V_{pi}=\sqrt{U_{RCA,i}}$ and the residual potential of the test mass $V_{TM}$ is assumed too small to neglect after allocation.

The knapsack problem is a typical optimization problem in operational research and many scholars have been attached to research for the problem. A thorough survey of the literature on the knapsack problem has recently been presented in [31,32]. Considering the dimension of the optimization variables is not high, a dynamic programming algorithm [33] is chosen to solve the problem of Equation (43).

Then, let
(44)$$V_{si}=\left\{\begin{array}{cc} V_{pi} & if\;f_{i}=1\\ -V_{pi} & if\;f_{i}=0 \end{array}\right..\;$$

After this step, the residual potential of the test mass is
(45)$$V_{rTM}=\frac{\sum_{i=1}^{n}C_{i,TM}V_{si}}{\sum_{i=1}^{n}C_{i,TM}}.\;$$

Compensating the effect of $V_{rTM}$, the actual potential of electrodes is given as follows:(46)$$V_{i}=V_{si}+V_{rTM}.\;$$

To sum up, the complete flow chart of the proposed robust control allocation algorithm is shown in Figure 4.

## 4. Simulations

To study the effectiveness and the performance of the proposed robust control al-location method, numerical simulations have been carried out for the space inertial sensor under test mass release phase with overcritical conditions. All the computations and plots are performed using the MATLAB/SIMULINK software package, and MO-SEK Toolbox is used to solve the programming problems.

Twelve pairs of electrodes are mounted in the space inertial sensor to reliably control the position and attitude motion of the test mass. Figure 5 shows the schematic configuration of those electrodes.

The drag-free spacecraft contains and shields free-floating test mass from all non-gravitational disturbing forces. The disturbances acted on the spacecraft are measured by the inertial sensor and further offset by the control force from the Field Emission Electric Propulsion (FEEP). There is no contact between the spacecraft and the test mass, so the influence of disturbances acting on the test mass can be ignored. In the simulation, the collisions between the test mass and the electrode house are as-sumed to have occurred, resulting in large uncertainties in position measurements, at-titude measurements and control voltages. These uncertainties are assumed to be 100 times of that required by the LISA Pathfinder mission as listed in Table 1 [34]. Since the main focus of this section is to verify the control allocation algorithm, the widely used proportional-derivative controller (PD) controller is used as the motion control algo-rithm to generate commanded electrostatic forces and torques. The parameters of the simulations presented in the section are provided in Table 2. The traditional pseu-do-inverse control allocation method is also studied in this section for comparison. For the sake of clarity, the proposed robust control allocation method is labeled as “Rob-CA”, and the pseudo-inverse control allocation method is labeled as “PICA”.

**Remark** **1.**
*The existence of non-diagonal elements of inertia moments would cause cross-coupling actuation noise, which will result in a significant reduction in measurement accuracy in scientific measurement mode. To avoid cross-axis coupling effects, the test mass is designed as a standard cube. By strictly controlling the fabrication process, the non-diagonal elements of inertia moments of test mass are small enough to be ignored.*


Figure 6 and Figure 7 show the time response comparisons of position and attitude, and velocity and angular velocity of the test mass under the aforementioned PD controller with the proposed RobCA method and the PICA method, respectively. It can be seen that under the RobCA method, the test mass reaches the commanded target with a settling time less than 50 s, while it is more than 90 s under the PICA method. Moreover, the oscillation amplitude of the states of the test mass under the RobCA method is also smaller than that of the PICA method. In addition, since the weight function of the RobCA method unifies the control forces and torques to the same scale, the position and attitude of the test mass reach the commanded target almost simultaneously.

Figure 8 presents the trajectories of the position ${\left[\begin{array}{ccc} x & y & z \end{array}\right]}^{T}$ and attitude ${\left[\begin{array}{ccc} \varphi & \theta & \psi \end{array}\right]}^{T}$ of the test mass with the RobCA method and the PICA method. It is clearly described that the trajectories of the RobCA method are smoother and shorter than that of the PICA method, which further illustrates the effectiveness of the RobCA method.

The comparison of the nearest distance between the test mass and the electrode house along *x*, *y*, *z* axis with the RobCA method and the PICA method are presented in Figure 9. Figure 9 shows that the nearest distance between the test mass and the electrode house with the RobCA method is almost always larger than that of the PICA method. Accordingly, there is a greater safety margin in the test mass capture phase while the RobCA method is used as the control allocation method.

Define the error between the commanded forces and torques ***v_d_*** and actual electrostatic forces and torques after control allocation ***v*** as control allocation error, i.e., $v_{e}=v_{d}-v$. The time response comparison of the control allocation error is shown in Figure 10. It is clear that the control allocation error of the RobCA method is less than that of the PICA method. The average norm of forces allocation error and torques allocation error with the RobCA method are 5.5 × 10^−6^ and 6.2 × 10^−8^, respectively, which are only 79% and 43% of that of the PICA method.

The time response of the residual voltages of the test mass is shown in Figure 11. It can be seen that the residual voltages of the test mass are less than 1 V throughout the simulation, which can be acceptable.

## 5. Conclusions

This paper has addressed the 6-DOF stabilization problem of a test mass of the drag-free spacecraft’s space inertial sensor with state measurement uncertainty and control voltage uncertainty under test mass release phase with overcritical conditions. A 6-DOF test mass dynamics model with uncertainty is established, and the influence of uncertainty on control allocation is analyzed. A robust control allocation method is proposed to suitably distribute the commanded forces and torques into individual electrodes. Numerical simulations demonstrate the effectiveness and advantages of the proposed control allocation scheme. The simulation results show that compared with the traditional pseudo-inverse control allocation method, the proposed robust control allocation method can achieve a shorter settling time, greater security margin and smaller allocation error. However, the proposed methods only provide the synthetic uncertainty of the system, but cannot identify the individual uncertainty of the actuators, which are the subjects of our future research.

## Figures and Tables

**Figure 1 sensors-23-02881-f001:**
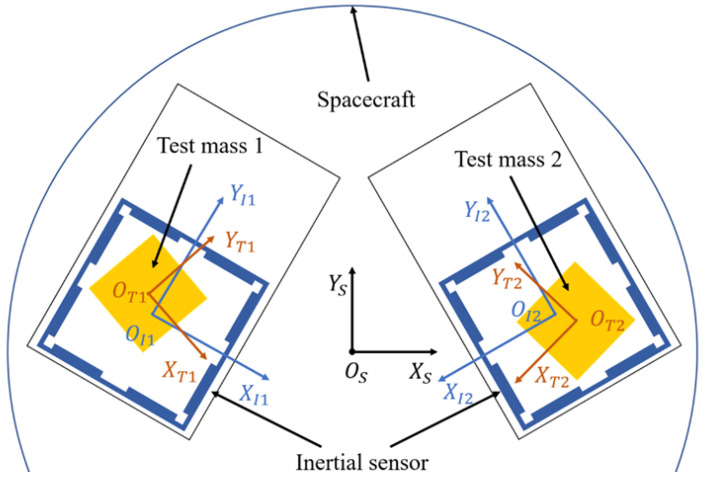
The simplified relative motion coordinate system for drag free spacecraft.

**Figure 2 sensors-23-02881-f002:**
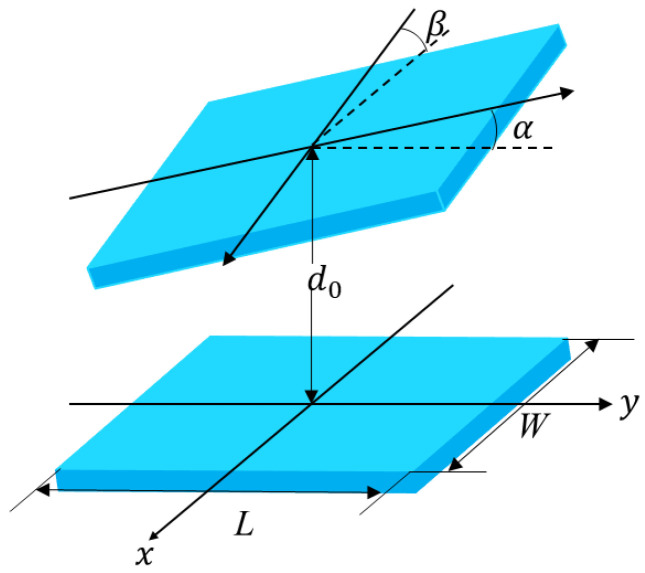
Diagram of two electrodes with relative rotation.

**Figure 3 sensors-23-02881-f003:**
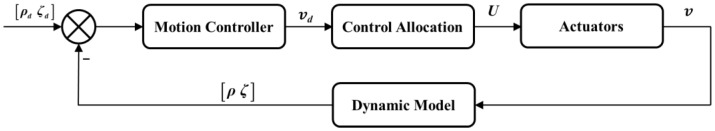
Structure of control systems with redundant actuators.

**Figure 4 sensors-23-02881-f004:**
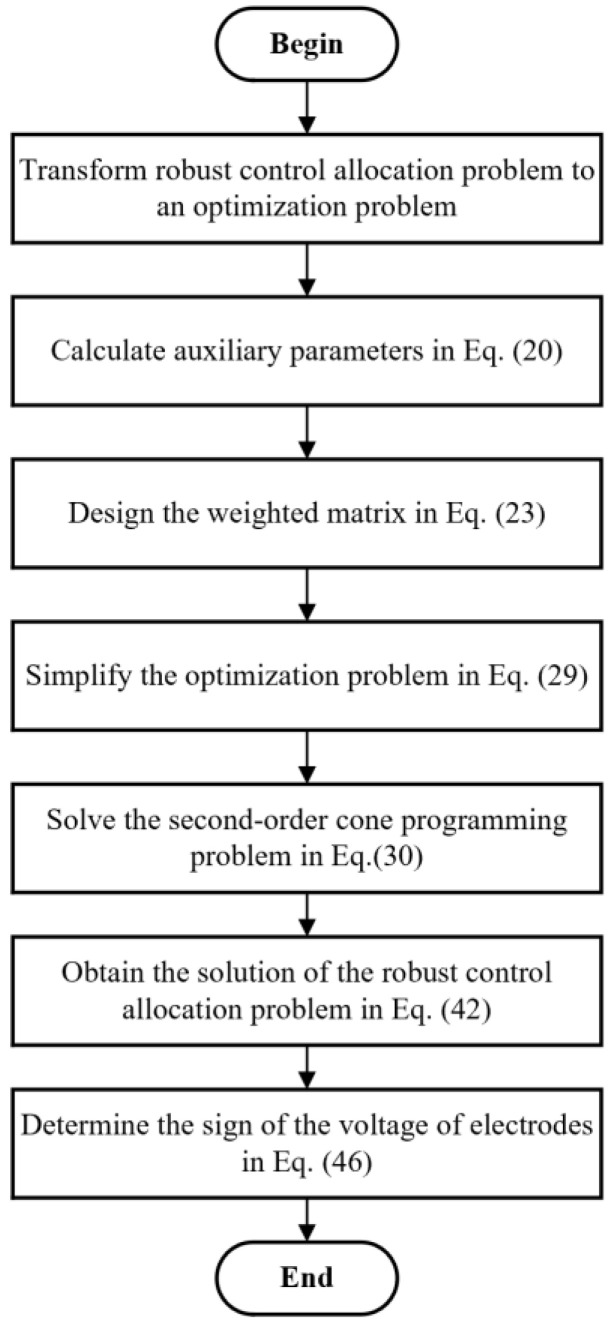
Flow chart of the proposed robust control allocation algorithm.

**Figure 5 sensors-23-02881-f005:**
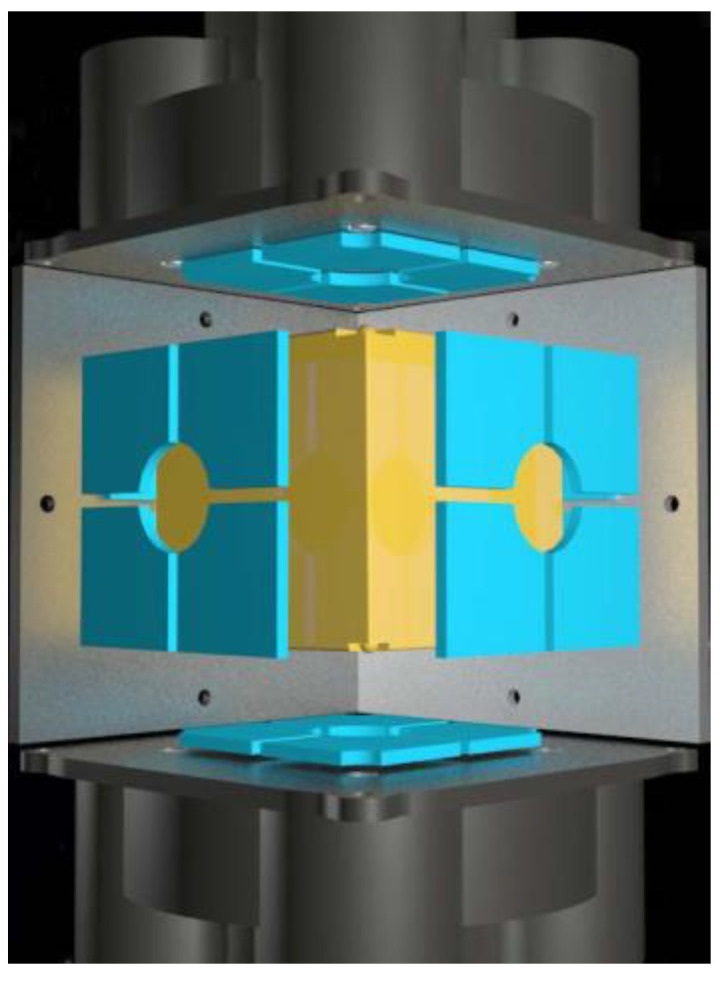
Distribution schematics of the 12 pairs of electrodes.

**Figure 6 sensors-23-02881-f006:**
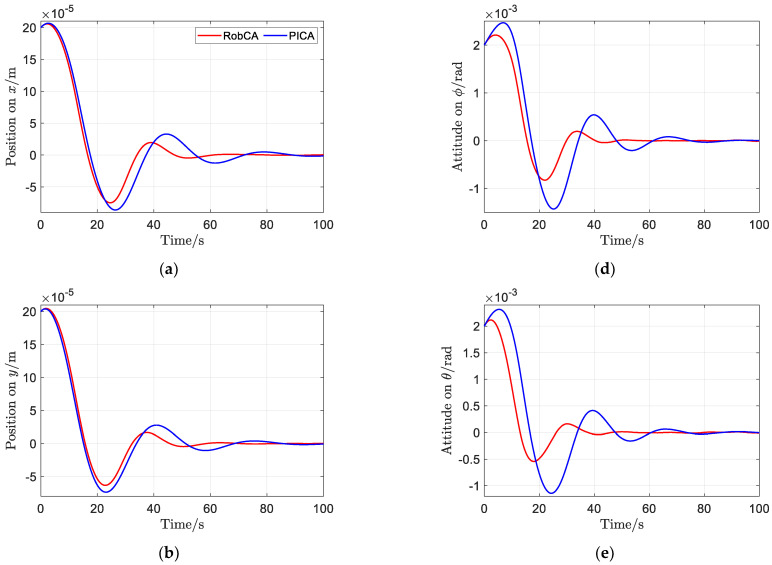
Time response comparisons of position and attitude of the test mass with the proposed RobCA method and the PICA method. (**a**) Position on *x*. (**b**) Position on *y*. (**c**) Position on *z*. (**d**) Attitude on *ϕ*. (**e**) Attitude on *θ*. (**f**) Attitude on *ψ*.

**Figure 7 sensors-23-02881-f007:**
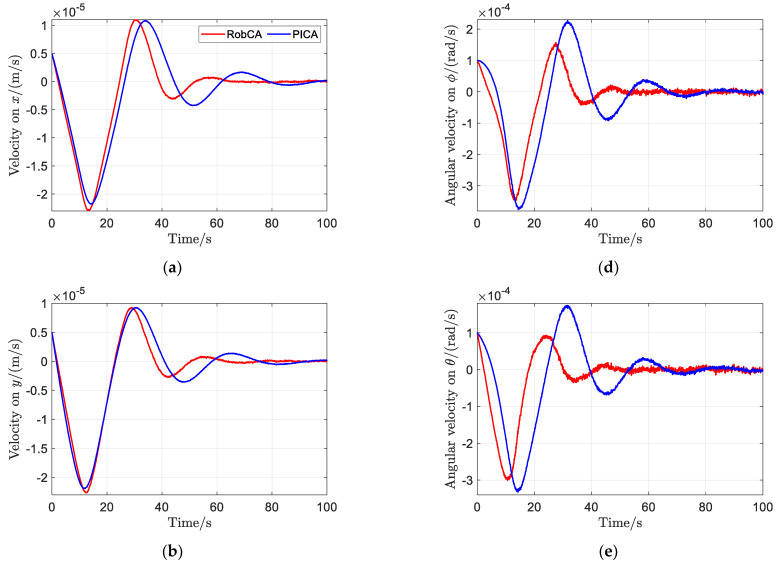
Time response comparisons of velocity and angular velocity of the test mass with the proposed RobCA method and the PICA method. (**a**) Velocity on x. (**b**) Velocity on y. (**c**) Velocity on z. (**d**) Angular velocity on ϕ. (**e**) Angular velocity on θ. (**f**) Angular velocity on ψ.

**Figure 8 sensors-23-02881-f008:**
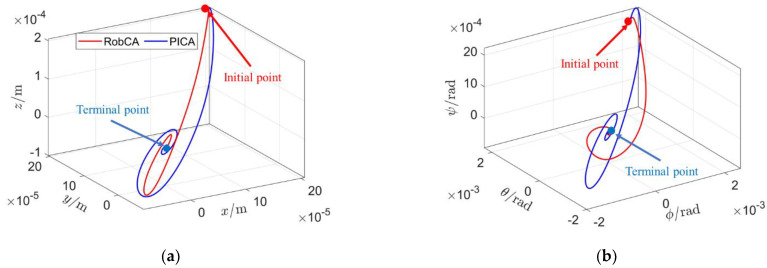
Trajectories of the position and attitude of the test mass with the RobCA method and the PICA method. (**a**) Position trajectories. (**b**) Attitude trajectories.

**Figure 9 sensors-23-02881-f009:**
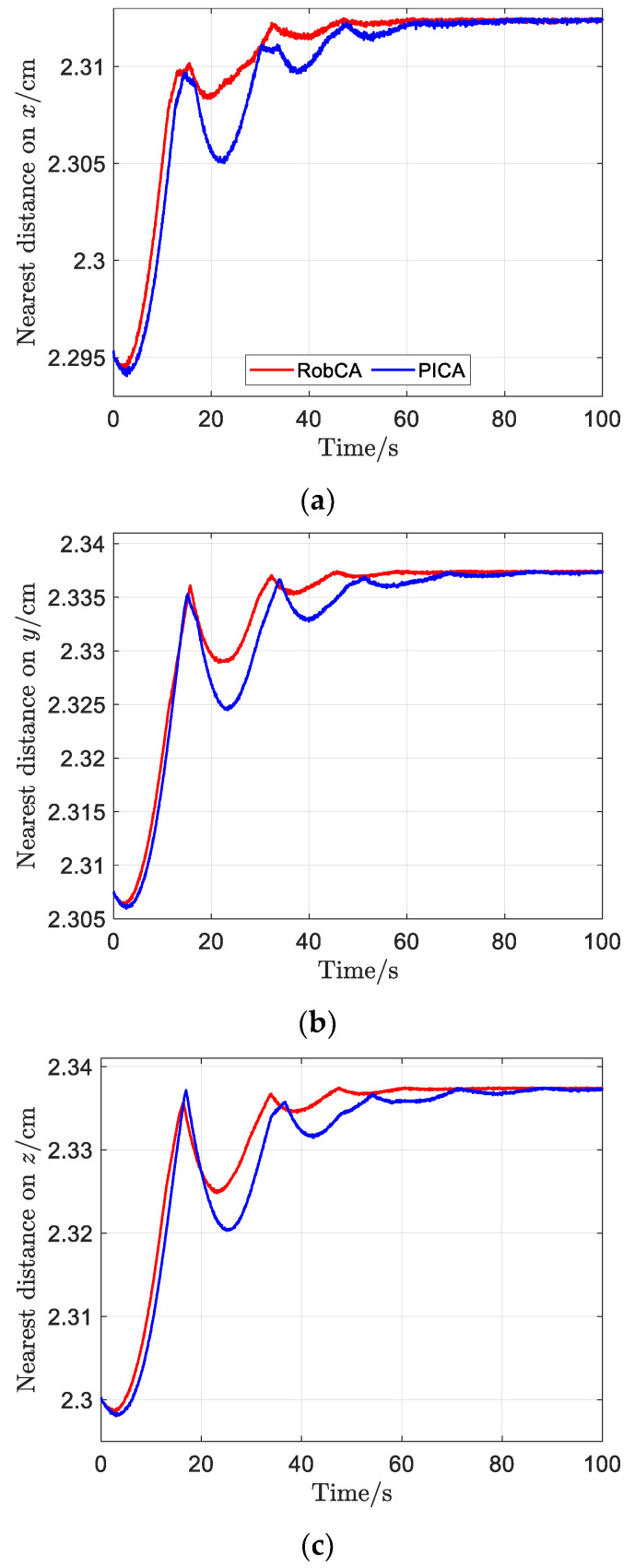
The nearest distance between the test mass and the electrode house with the RobCA method and the PICA method. (**a**) Nearest distance on x. (**b**) Nearest distance on y. (**c**) Nearest distance on z.

**Figure 10 sensors-23-02881-f010:**
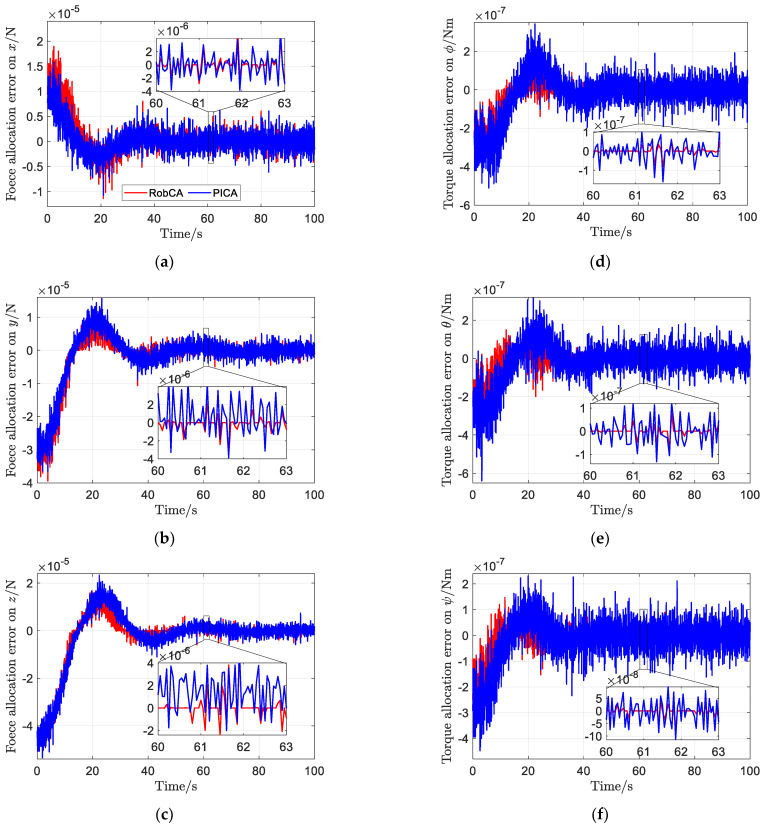
The nearest distance between the test mass and the electrode house with the RobCA method and the PICA method. (**a**) Force allocation error on x. (**b**) Force allocation error on *y*. (**c**) Force allocation error on *z*. (**d**) Torque allocation error on ϕ. (**e**) Torque allocation error on *θ*. (**f**) Torque allocation error on *ψ*.

**Figure 11 sensors-23-02881-f011:**
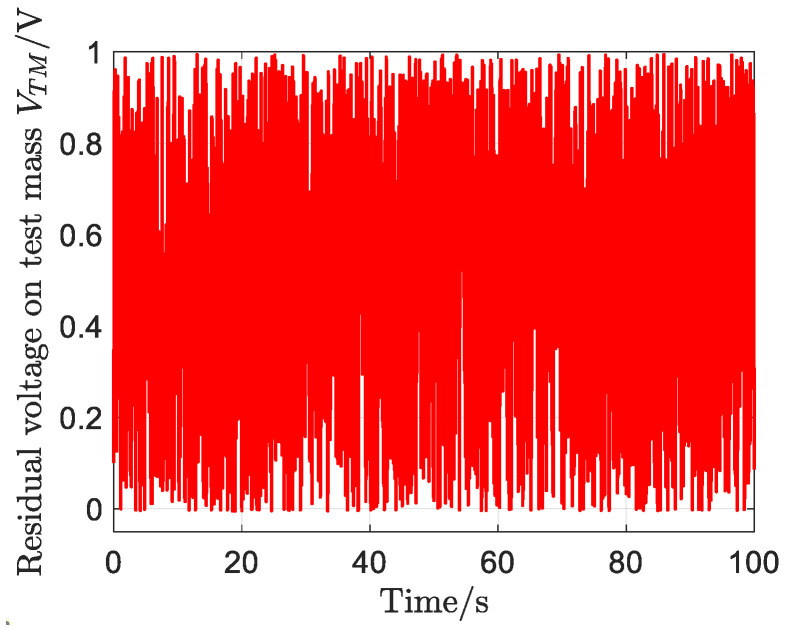
Time response of the residual voltages of the test mass.

**Table 1 sensors-23-02881-t001:** The initial release states of the test mass in the IBF frame and the uncertainties in the control system.

Parameter	Value
Initial release position	$\rho_{0}=\left[\begin{array}{ccc} 2 & 2 & 2 \end{array}\right]\times 10^{-4}m$
Initial release velocity	$v_{0}=\left[\begin{array}{ccc} 5 & 5 & 5 \end{array}\right]\times 10^{-6}m/s$
Initial release attitude	$\left[\begin{array}{ccc} \varphi & \theta & \psi \end{array}\right]=\left[\begin{array}{ccc} 2 & 2 & 2 \end{array}\right]\times 10^{-3}\mathrm{rad}$
Initial release angular velocity	$\omega_{0}=\left[\begin{array}{ccc} 1 & 1 & 1 \end{array}\right]\times 10^{-4}\mathrm{rad}/s$
Uncertainty of position measurement	$d_{mp}=1.7\times 10^{-5}m/\sqrt{\mathrm{Hz}}$
Uncertainty of attitude measurement	$d_{ma}=2\times 10^{-5}\mathrm{rad}/\sqrt{\mathrm{Hz}}$
Uncertainty of control voltage	$d_{V}=1\times 10^{-4}V/\sqrt{\mathrm{Hz}}$

**Table 2 sensors-23-02881-t002:** The parameters of the numerical simulations.

Parameter	Value
Mass of test mass	$m_{tm}=2.45\mathrm{kg}$
Inertia moments of test mass	$I_{tm}=diag\left\{1,\;1,\;1\right\}\times 10^{-3}\mathrm{kg}\cdot m^{2}$
Position of the spacecraft in the frame ECI	$r_{s}=\left[\begin{array}{ccc} -8.614 & -4.629 & -2.065 \end{array}\right]\times 10^{7}m$
Position vector from the geometric left of space inertial sensor to spacecraft	$r_{is}=\left[\begin{array}{ccc} 0.3 & 0 & 0 \end{array}\right]m$
Orbital angular velocity of test mass	$\omega_{b}=\left[\begin{array}{ccc} 0 & 1.9975 & 0 \end{array}\right]\times 10^{-5}\mathrm{rad}/s$
Length of test mass	$l_{tm}=0.05\;m$
Center distance of adjacent electrodes	$L_{\varphi }=L_{\theta }=L_{\psi }=0.03\;m$
Area of each electrodes	$A=4.05\times 10^{-4}m^{2}$
Ideal distance between test mass and electrodes	$d_{io}=\left[\begin{array}{ccc} 5 & 4 & 4 \end{array}\right]\times 10^{-3}m$
Limit of electrodes voltage	$-100\;V\leq V_{i}\leq 100\;V$
Controller parameters in position	$K_{p}=0.075,K_{d}=1.85$
Controller parameters in attitude	$K_{p}=0.125,K_{d}=3.75$
Constant *η* in Equation (13)	$\eta =0.05\left\|B\right\|$
Constant *κ* in Equation (20)	κ=0.01

## Data Availability

Not applicable.

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
