# Peer review of "Robust Control Allocation for Space Inertial Sensor under Test Mass Release Phase with Overcritical Conditions"

_sensors, 2023, doi:10.3390/s23062881_

Round 1

Reviewer 1 Report

The paper is well written. It needs some minor revisions:

-Improve the abstract with more details and highlighted the achievements.

-Hope for more related references from MDPI publications.

Author Response

Many thanks for your time and patience in reviewing our manuscript. Attached please find the revised manuscript and response letter, which we would like to submit for your kind consideration.

Reviewer 2 Report

This paper has been written clearly and involves some interesting aspects. However, the following modifications are needed for recommending the paper for publication:

1. Transferring diagram 5 to the introduction section can ease understanding of the problem.

2. Line 209 “A control system with redundant actuators can usually be divided into two serial levels, which can be seen in Figure 2” Figure 2 is diagram of two electrodes with relative rotation, please check it.

3. Line 225, Line 257: Why the first letter is not uppercased?

4. Line 405: “Define the error between the commanded forces and torques τc and actual electrostatic forces and torques after control allocation ? as control allocation error, i.e., ve=vd-v”. Clarify the relationship between? and v?

5. Line 175 “is denoted as the initial distance between the point Pi to the center of the i-th electrode”. What is denoted? Maybe a symbol is missed.

Author Response

(The authors gave the same response as above.)

Reviewer 3 Report

The authors addressed a 6-DOF stabilization problem of a test mass of the drag-free spacecraft’s space inertial sensor with state measurement uncertainty and control voltage uncertainty under test mass release phase with overcritical conditions. The 6-DOF test mass dynamics model with uncertainty is established, and a robust control allocation method is proposed to suitably distribute the commanded forces and torques into individual electrodes. Numerical simulations demonstrate the effectiveness and advantages of the proposed control allocation scheme. Therefore, I think this work is interesting and readers can benefit from this paper. However, there are some problems which could be solved carefully before considering for publication.

1. How to deal with disturbances acted on the spacecraft while considering control allocation?

2. The inertia moments of test mass in the simulation should be with a more general case rather than a diagonal matrix.

3. Some editing remarks:

a) line 194: Eq. (10) ∂/ ∂qj=∂[εAj(di+∆di)], How did you come up with the equation? Please confirm it.

b) line 161, line 173, line250: the space at the beginning of each sentence should be deleted.

c) line 293: “Note that, if……” is the Imperative mood really needed?

Author Response

(The authors gave the same response as above.)
